# Experimental Investigation on Machinability of Aluminum Alloy during Dry Micro Cutting Process Using Helical Micro End Mills with Micro Textures

**DOI:** 10.3390/ma13204664

**Published:** 2020-10-19

**Authors:** Yao Sun, Liya Jin, Yadong Gong, Yang Qi, Huan Zhang, Zhipeng Su, Kai Sun

**Affiliations:** 1School of Mechanical Engineering and Automation, Northeastern University, Shenyang 110819, China; sy547515291@163.com (Y.S.); liyajin1997@163.com (L.J.); 117049325@163.com (Y.Q.); zhangh030600@163.com (H.Z.); zhipesu@163.com (Z.S.); 2Siasun Robot and Automation Co., Ltd., Shenyang 110169, China; sunkai@siasun.com

**Keywords:** aluminum alloy material, surface roughness, dry micro cutting, micro textures

## Abstract

Aluminum alloy material is widely used in the electronics, weapons, aviation and aerospace industries, due to its medium strength, good corrosion resistance, good toughness and excellent oxidation properties. With the trend of product miniaturization, micro cutting has become the mainstream technique for fabricating micro parts and components, so it is very meaningful and vital to work on removing the cutting fluid from the micro cutting process and make it totally sustainable and eco-friendly. In this work, an attempt has been made to fabricate micro textures onto the rear surface of helical micro end mills with diameters of less than 1 mm. Micro textures in the form of grooves were fabricated using a noncontact low speed wire electrical discharge turning technique. Dry micro cutting experiments were carried out on an aluminum alloy material using helical micro end mills with micro textures and the dry micro cutting surface quality and tool wear have been investigated. The influence of dry micro cutting parameters on the surface roughness parameters were also investigated. Experimental results showed that the *S*_a_ and *S*_q_ can be reduced to be about 1.56 μm and 2.08 μm, respectively. Contrasting results indicate that the implantation of micro textures does not deteriorate the dry micro cutting surface but improves the machined surface consistency of an aluminum alloy workpiece. The tool wear on helical micro end mills with micro textures involved in the dry micro cutting process of Al 6061 mainly include rear frictional wear, oxidation wear and diffusion wear.

## 1. Introduction

Micro cutting technology can meet the demand for precision micro parts and components with characteristic sizes ranging from micron to millimeter, extensive throughout the aerospace, national defense, biological engineering and precision instruments industries [1,2]. However, the micro cutting process has its own characteristics and rules, which are different from traditional cutting processes [3]. Due to the sharp reduction in tool size (the diameter of cutting tools involved in the micro cutting process is usually less than 1 mm), the size effect is obvious, together with the relatively large cutting force and heat generated in the process of micro cutting that aggravates rapid tool wear. Therefore, many researchers have adopted an analytical or experimental approach to disclose the processing mechanisms of micro cutting technology. Wojciechowski et al. [4] developed the cutting force predicted model considering the chip thickness accumulation phenomenon and they found that the cutting force’s nonlinearity was significantly affected by multiple cutting mechanism transitions. Wan et al. [5] proposed an analytical micro cutting force model considering the plastic formation and slip-line field theories and conducted experiments to verify the reasonability of this model. Chen et al. [6] established the uncut chip thickness model and an improved cutting force model of micro milling by considering the ploughing effect and elastic recovery of the workpiece material. Aurich et al. [7] used single edge micro end mills with diameter of 50 μm and investigated the effects of the title angle of the spindle on the surface quality in micro milling process by kinematic simulations and experiments. Liu et al. [8] disclosed that a smaller feed per tooth could make the tool material stick less easily to the titanium alloy in the micro milling process.

The dry micro cutting process as an eco-friendly technique without air and water pollution belongs to green cutting technology, which refers to entirely avoiding the use of cutting fluids and is a more sustainable method over MQL (minimum quantity lubrication) and conventional wet machining [9]. Muaz et al. [10] investigated the multiobjective optimization of AISI 4340 steel machined by MQL-assisted cutting processes and their results indicated that low viscosity fluids could improve lubrication performance. Qu et al. [11] found that the carbon nanofluid MQL conditions could obtain better surface quality and lower grinding forces compared with dry, flood and MQL conditions and further studied the effects of nanoparticle concentration and air pressure on grinding forces and surface quality. Sharma et al. [12] used an LB-6000 vegetable metalworking fluid-based MQL system during machining D2 steel and investigated the influences of MQL on surface roughness and cutting temperature. Ni et al. [13] disclosed that the surface roughness obtained with an ultrasonic machining method and MQL technique is improved by about 30–50% and about 20–30% compared with conventional milling and ultrasonic vibration-assisted milling, respectively. Maruda et al. [14] investigated the effects of emulsion mist cooling on the conditions of heat absorption from the cutting zone and found that the content of surface active compounds in the contact area was much larger compared with the areas beyond the contact area. However, there are some drawbacks like high friction and adhesion involved in dry micro cutting processes that can reduce the service life of micro cutting tools, so many efforts have been made to develop new tool material, optimize tool geometry design and adopt tool coating techniques for alleviating the drawbacks of the dry machining process [15].

In recent years, surface texturing technology has been found effective to reduce the tool and chip contact area and store lubricant, so various micro textured structures have been introduced to rake and rear the surface of cutting tools or grinding wheels [16]. Sugihara et al. [17,18] pointed out that the implantation of striped micro textures on the tool surface can significantly improve the antiadhesion of the tool and have strong wear resistance in the metal cutting process. Obikawa et al. [19] fabricated four kinds of cutting tools with groove and ridge textures and various widths, depths and distances from the cutting edge and found that parallel and dot type textured tools are conducive to reduce friction coefficient as well as cutting forces. Xie et al. [20] used a V-shaped diamond wheel to fabricate precise and smooth micro grooves on the surface of lathe tools and then carried out dry turning experiments, their study results showed that the micro grooves formed on the front face of the tool were helpful to reduce the chip friction and eliminate cutting heat. Fatima et al. [21] fabricated grooved micro textures on the front and back surface of a carbide turning tool, the orthogonal experiments of AISI4140 steel were carried out under dry cutting conditions, and their results indicated that the friction coefficient was decreased by 17 and 18% compared to tools without micro textures. Kummel et al. [22] used laser machining to manufacture the micro pits, micro textures and micro grooves on the front surface of cemented carbide tools to reduce the adhesion of the tool when cutting 1045 steel and increase the stability of the cutting edge. Rathod et al. [23] found that textured tools could reduce cutting force and adhesion behavior of working materials more effectively compared with nontextured tools, and in terms of cutting force reduction and surface finish improvement, the new square textured tool was superior to the linear textured tool. Arulkirubakaran et al. [24] fabricated different kinds of micro texture onto a cemented carbide tool, including parallel or perpendicular to the cutting edge and cross textures, and their results disclosed that the vertical cutting edge textured tool had the least cutting force and the least friction coefficient in the machining process of titanium alloy. Kumar et al. [25] revealed that micro texture at an angle to chip flow can effectively reduce friction coefficient and tool stress.

From literature, many studies paid attention to the fabrication of different micro textural patterns on the rake or rear surface of a turning tool, or conventional sized milling cutter, etc., which has proved to be an effective approach to significantly improve surface quality and tool wear in the dry cutting process. However, there are few studies concerning the fabrication of micro textures on micro cutting tools with a diameter of less than 1 mm and what is more, even fewer studies focused on investigating the dry micro cutting performance of helical micro end mills with micro textures. Therefore, to fill this gap, this study paid attention to developing a helical micro end mill with micro grooves located on the flank face using the EDM (electrical discharge machining) method, and the influences of micro dry cutting parameters on the surface quality and micro cutting forces of aluminum alloy 6061 have been thoroughly investigated. Furthermore, the dry micro cutting mechanisms of micro helical end mills with micro textures are disclosed based on the surface morphology and micro cutting force.

## 2. Materials and Methods

### 2.1. Preparation of MT-HMEM with EDM Method

The EDM process removes material by pulsed spark discharge, is not restricted by the strength and hardness of materials, and it does not have direct physical contact with micro cutting tools, which is different from turning or grinding methods, so the noncontact machining nature makes it more competent to fabricate micro cutting tools [26,27,28]. In this study, the LS-WEDM (low speed wire electrical discharge machining) tool CA20 (Beijing, China) made by Beijing Agiecharmilles combining with the precise rotating unit was applied to fabricate two groups of micro end mills, namely MT-HMEM (helical micro end mills with micro texture) and HMEM (helical micro end mills). Environmentally friendly deionized water was used as the working fluid instead of polluting kerosene with conductivity of 5 μS/cm and temperature of 25 °C, and brass wire with the diameter of 0.2 mm was used as the tool electrode. Firstly, the helical micro end mill substrate with diameter of about 800 μm was reduced from the original rod with a diameter of 2 mm; then, the helical blades were obtained with selected rotating speed and feeding speed; finally, the micro structures were formed on the rear surface of helical micro end mills as presented in Figure 1. The specific machining parameters for fabricating MT-HMEM are displayed in Table 1.

The fabrication results of helical micro end mills with or without micro textures were measured using the VHX-1000E microscope (Keyence, Osaka, Japan) as presented in Figure 2a,b, respectively. The measured results indicated that the diameter of MT-HMEM was about 800 μm, the axial cutting edge length was 2 mm and the helical angle was about 30°. The local surface morphology and surface element were investigated as in Figure 2c,d, the surface was characterized with discharge craters, micro cracks, and droplets. The base material of MT-HMEM was a hard alloy (YG8) consisting of Co and W elements, and the detected results indicated that on the MT-HMEM surface there existed foreign elements including O, Cu and Zn. The three dimensional view of MT-HMEM presented that the groove distance was about 150 μm and the depth was about 8 μm as shown in Figure 2e. If the shape and size of the micro textures are the same and evenly distributed, the mathematical model of the micro texture surface for MT-HMEM can be deduced, based on sine formula as Equation (1).
(1)$$z=\left\{ \begin{array}{l} c_{1}sin\frac{\pi }{\lambda_{1}}x,0\leq x\leq \lambda_{1}\\ c_{2}sin\frac{\pi }{\lambda_{2}}\left(x-\lambda_{1}\right),\lambda_{1}\leq x\leq \left(\lambda_{1}+\lambda_{2}\right) \end{array}\right.$$
where *x* = 0,1,2,… and represents the sequence number of the micro textures distribution in the x direction; *c*_1_ and *c*_2_ respectively are the height of wave crest and wave trough; *λ*_1_ and *λ*_2_ are the wavelengths of single wave crest and wave trough.

### 2.2. Design of Dry Micro Cutting Experiments

The dry micro cutting experiments were performed on JX-1A precision micromachining tool (Juxie, Beijing, China) as displayed in Figure 3, which was composed of a spindle, vacuum chucks, coolant and digital control system, and its position resolution was 0.1 μm. The Al 6061 as the typical viscous material was selected in this experiment. It is widely used in the field of aerospace and its chemical properties are displayed in Table 2 [22]. The micro cutting tool was a helical micro end mill with micro textures made of YG8 material as displayed in Figure 3. Considering the small diameter and minimum cutting thickness of MT-HMEM tool, the experiment of micro cutting was carried out with high spindle speed, small feed rate and small radial cutting depth. The single factor experiment was designed and the rotating speed *V_n_*, feed speed *V_f_* and micro cutting depth *a_p_* were selected as listed in Table 3, and axial tangential depth was 0.4 mm.

The surface microstructure and foreign elements of the machined surface were characterized using scanning electron microscopy and energy dispersive spectroscopy (EDS) (Zeiss, Oberkochen, Germany) analysis. The confocal laser scanning microscope (Olympus, Tokyo, Japan) made by OLYMPS (the shear stress on the shear plane) was used to measure the surface roughness parameters including the *S*_a_ (arithmetic mean height) and *S*_q_ (root mean square height), which differed from the line surface roughness parameters *R*_a_ and *R*_q_, and the tool wear of MT-HMEM and HMEM after machining was also observed. The force signal was collected by ME FC-K3D60 dynamometer with full range interference accuracy of 0.5% and the range of ±20N in three directions.

Additionally, the schematic diagram of the dry micro cutting process using the MT-HMEM tool is presented in Figure 4, and the *dF_n_* and *dF_t_* of the dry micro cutting force for MT-HMEM can be deduced as Equation (2). Based on the geometric relationship in Figure 4, Equation (3) can be obtained. According to Equation (2), it can be indicated that the *dF_n_* and *dF_t_* are proportional to the contact length, so the tangential force (*F_t_*) and normal force (*F_n_*) of MT-HMEM are smaller than that of HMEM.

(2)$$\left\{ \begin{array}{cc} \; & dF_{t}=\left[\sigma 1lABsin\varphi +\tau 1lABcos\varphi -l_{BC}^{{}^{\prime }}\sigma 2\left(sin\beta -\tau 2cos\beta /\sigma 2\right)\right]dw\\ \; & dF_{n}=\left[\sigma 1lABcos\varphi -\tau 1lABsin\varphi +\sigma 2l_{BC}^{{}^{\prime }}\left(cos\beta +\tau 2sin\beta /\sigma 2\right)\right]dw \end{array}\right.$$(3)$$l_{BC}^{\prime }=l_{BC}-nl$$
where *d_w_* is the thickness of a micro blade unit; *σ*_1_ and *τ*_1_ are the normal stress and shear stress on the shear plane, respectively; the *σ*_2_ and *τ*_2_ are the normal stress and shear stress of the elastic recovery area of the rear blade under the action of ploughing, respectively; *φ* is shear angle; *β* is the rear angle of MT-HMEM; the *l′_BC_* is the contact length of MT-HMEM; *l_BC_* is the contact length of HMEM; *l*_1_ is the width of micro groove textures and *n* is the number of micro textures.

## 3. Results

### 3.1. Surface Microstructure of Micro Cutting Process

The surface microstructure and foreign elements of the dry micro cutting surface obtained with MT-HMEM and HMEM were investigated as shown in Figure 5 and Figure 6. Micro cracks could be observed on the surface machined by MT-HMEM which were related to the vibration of intermittent cutting caused by grooved micro textures on the MT-HMEM surface, but the dry micro cutting surface machined by HMEM was lacking in micro cracks, as presented in Figure 5a–d. Further, the foreign elements analysis results disclosed that the amount of O and C elements existing on the dry micro cutting surface was more than that on the surface machined with HMEM, as displayed in Figure 5f. This was due to the secondary micro cutting phenomenon generated by MT-HMEM and making more O elements from the oxidation reaction and C elements from the hard alloy MT-HMEM being deposited on the dry micro cutting surface.

### 3.2. Effects of Dry Micro Cutting Parameters on S_a_ and S_q_

Surface roughness has an important influence on the service life, fatigue life and work efficiency of a machined workpiece and is influenced itself by the workpiece’s material properties, chatter, machining parameters, etc., [30,31,32,33]. The effects of dry micro cutting parameters on three-dimensional surface roughness parameters, including *S*_a_ and *S*_q_, were investigated. The *S*_a_ and *S*_q_ decreased with the increase of *V*_n_ as presented in Figure 7a, and the *S*_a_ was reduced to about 1.56 μm when the rotating speed was 50,000 r/min. The *S*_a_ firstly decreased and then increased continuously with the feeding speed increase and the turning point of *V*_f_ was 40 µm/s as shown in Figure 7c. This was due to the small *V*_f_ corresponding to a small feed per tooth which resulted in plastic removal not occurring on the micro cutting surface obtained by MT-HMEM, but plowing and scratching predominated, resulting in a poor surface. When the *V*_f_ was increased, the material removal per unit time and the surface residue volume were increased which deteriorated the dry micro cutting surface of Al 6061 machined by MT-HMEM. Additionally, the effects of *a_p_* on *S*_a_ and *S*_q_ are displayed in Figure 7c, with the increase of *a_p_*, the *S*_a_ and *S*_q_ first decreased and then increased; When *a_p_* was smaller than 10 µm, plastic removal coexisted with plowing which caused the unstable cutting; When the *a_p_* was 10 µm, and the dry micro cutting surface was stable and producing continuous chip, the *S*_a_ and *S*_q_ were smallest; when the *a_p_* was further increased, and the per unit time of micro cutting volume was increased, the Al 6061 was sticky, which made it easy for the chip to pile up on the cutting edge and micro textures of MT-HMEM, resulting in elastic recovery and deteriorating dry micro cutting quality and increased surface roughness parameters. The *S*_a_ obtained with MT-HMEM and HMEM with various *a_p_* are compared in Figure 7d, the average *S*_a_ of Al 6061 machined by MT-HMEM and HMEM, respectively, were 2.207 µm and 2.171 µm, which indicated that the implantation of micro textures did not deteriorate the dry micro cutting surface compared with ordinary micro end mills.

### 3.3. Contrast of Dry Micro Cutting Surface Machined by MT-HMEM and HMEM

The dry micro cutting surfaces of Al 6061 machined by MT-HMEM and HMEM with *V*_n_ = 30,000 r/min, *V*_f_ = 60 μm/s and *a_p_* = 12 μm were compared as presented in Figure 8. It can be seen that the direction and pattern of surface topography undulation of Al 6061 machined by MT-HMEM and HMEM are obviously different. The gully obtained by MT-HMEM was very shallow and the material uplift direction was perpendicular to the feed direction, but the surface fluctuation of Al 6061 obtained with HMEM was parallel to the feed direction as shown in Figure 8b,d. This is due to the micro textures located on the rear surface of MT-HMEM involved in the cutting process which almost completely neutralized the surface ridges left by the helical blade.

The dry micro cutting surface generative mechanism for the various directions obtained with MT-HMEM and HMEM were investigated, and five surface contour lines were collected in transverse and longitudinal directions as presented in Figure 9a,b. The results showed that the *l*_tf_ (transverse contour fluctuation) of the dry micro cutting Al 6061 surface was larger than *l*_lf_ (longitudinal contour fluctuation) both for HMEM and MT-HMEM. Furthermore, the *l*_tf_ and *l*_lf_ obtained with HMEM could be confirmed to be 12.5 µm and 5 µm, respectively, which were smaller than those of the surface machined by MT-HMEM with 15 µm and 12.5 µm, as presented in Figure 9. The difference between the transverse and longitudinal contour fluctuations of the surface machined by MT-HMEM was 2.5 µm, which was much smaller than that of HMEM with 7.5 µm, indicating that the novel MT-HMEM was conducive to ensuring the surface consistency of Al 6061 in a dry micro cutting process.

### 3.4. Evaluation of Tool Wear of MT-HMEM after Dry Micro Cutting

The tool wear of micro cutting tools is small compared with conventional cutting tools, but in terms of its characteristic scale (the diameter of a micro cutting tool is usually smaller than 1 mm) and the machining precision of micro parts, tool wear will cause large machining dimensional errors and seriously affect the micro cutting force, temperature, surface integrity and service life of micro cutting tools. Thus, the tool wear of helical micro end mills with or without micro textures was observed and investigated as displayed in Figure 10. The HMEM, after dry micro cutting, revealed an obvious breakage of the tool nose, but the MT-HMEM did not suffer this phenomenon as displayed in Figure 10b,d. This was due to the rake and rear face of the micro cutting tool and workpiece material constantly encountering extrusion and friction in the dry micro cutting process of the aluminum alloy, which would cause the rise of temperature and stiffness degradation of MT-HMEM and HMEM, together with small machining parameters and the high spindle speed adopted in the dry micro cutting process, resulting in high temperature and stress concentrated near the tool nose, but the existence of the micro textures on the MT-HMEM contributed to an enhanced heat dissipation effect and improved tool wear, so the MT-HMEM did not present tool nose breakage. Both the MT-HMEM and HMEM showed frictional flank wear, which was related to the constant contact and friction between the rear surface of micro end mills and the machined surface arising from the small cutting thickness.

Moreover, the cross profile of MT-HMEM indicated that the micro textures incurred evident wear and tear as presented in Figure 11, and the line scanning analysis result showed that the Al and O elements could be detected on the hard alloy MT-HMEM as displayed in Figure 11b–g. Some of the Al elements came from Al 6061, due to its strong dispersal potential, and the micro grooved texture could store some debris resulting in the increase of the content of Al; the WC and Co elements of the hard alloy reacted with oxygen in the air to form an oxidation film with low hardness in the dry micro cutting process resulting in the existence of O. Thus, the tool wear mainly included rear frictional wear, oxidation wear and diffusion wear for MT-HMEM when fabricating Al 6061 material without cutting fluids.

## 4. Conclusions

The present research attempted to investigate the machinability of aluminum alloy machined by micro end mills with micro textures in a dry micro cutting process. Firstly, the MT-HMEM made of hard alloy with a diameter of about 800 μm, the axial cutting edge length of 2 mm and helical angle of about 30° was successfully fabricated by the EDM method. Then the dry micro cutting experiments of MT-HMEM and HMEM were carried out and experimental results revealed that the implantation of micro textures not only deteriorated the dry micro cutting surface roughness but also improved the surface consistency due to the micro textures located in the rear surface of MT-HMEM involved in the cutting process almost completely neutralizing the surface ridges left by the helical blade. Finally, the line scanning analysis result disclosed that foreign elements, including Al and O could be detected on the MT-HMEM, and furthermore, the MT-HMEM after dry micro cutting did not show obvious breakage of the tool nose which revealed a remarkable improvement in the MT-HMEM in its resistance to tool wear. This study clearly revealed the superiority of using helical micro end mills with a micro texture for fabricating sticky aluminum alloy materials compared to ordinary helical micro end mills, in terms of tool wear and machined surface consistency.

## Figures and Tables

**Figure 1 materials-13-04664-f001:**
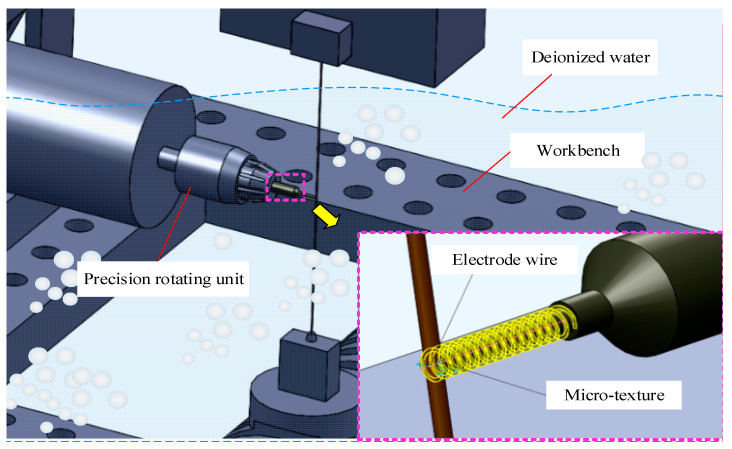
Preparation process of helical micro end mills with micro texture (MT-HMEM) using a low speed wire electrical discharge machining (LS-WEDM) method with deionized water.

**Figure 2 materials-13-04664-f002:**
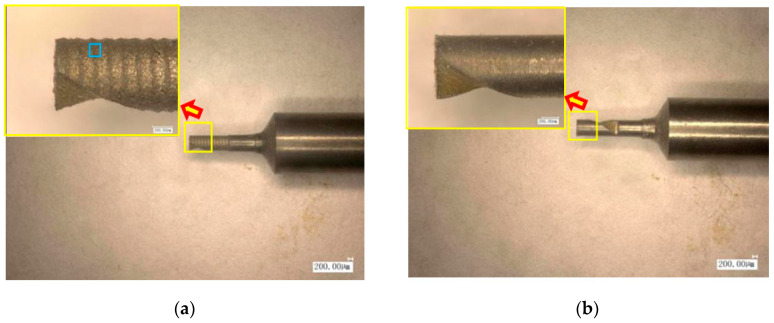
Fabrication results of MT-HMEM and HMEM (**a**) MT-HMEM, (**b**) HMEM, (**c**) local enlarged view of MT-HMEM, (**d**) EDS analysis, (**e**) three-dimensional view.

**Figure 3 materials-13-04664-f003:**
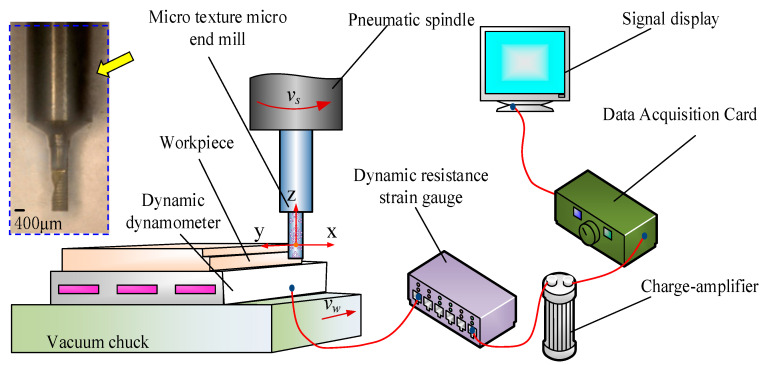
Dry micro cutting experimental and detection process.

**Figure 4 materials-13-04664-f004:**
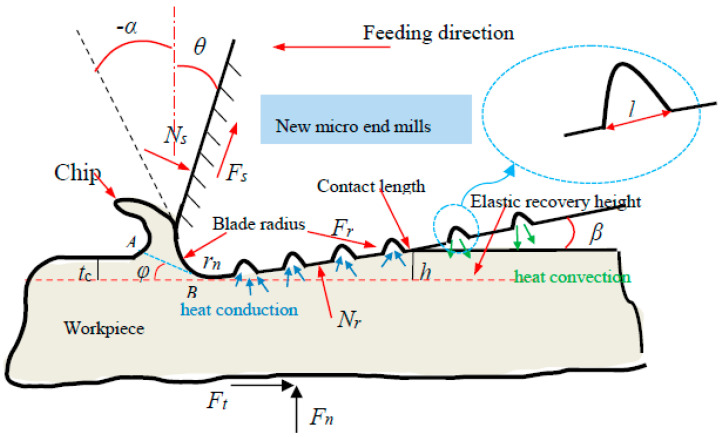
The schematic diagram of dry micro cutting process with MT-HMEM.

**Figure 5 materials-13-04664-f005:**
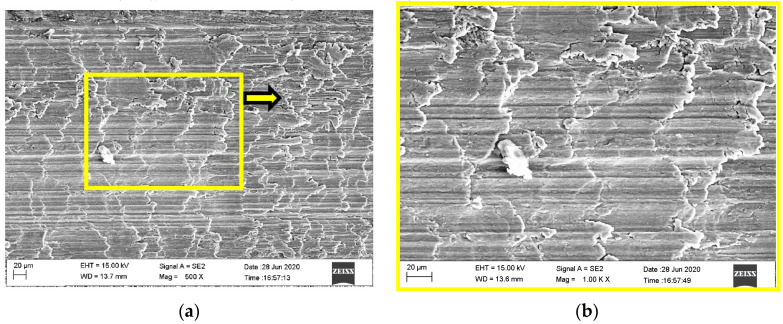
Micro cutting surface microstructure and foreign elements machined by MT-HMEM with *V*_n_ = 30,000 r/min, *V*_f_ = 40 μm/s and *a*_p_ = 12 μm; (**a**) surface microstructure; (**b**) local amplification; (**c**) cracks; (**d**) protrusion; (**e**) the EDS analysis of position A; (**f**) the EDS analysis of position B.

**Figure 6 materials-13-04664-f006:**
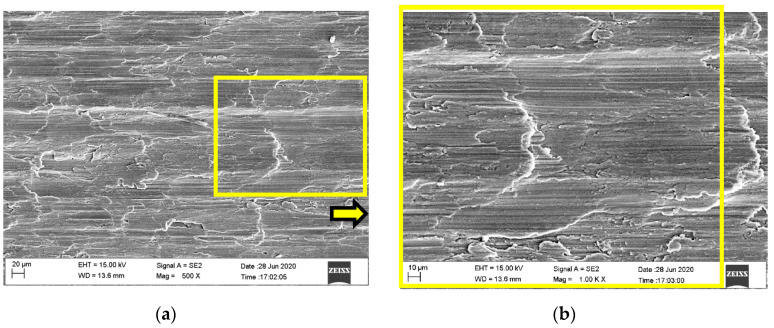
Micro cutting surface microstructure and foreign elements machined by HMEM with *V*_n_ = 30,000 r/min, *V*_f_ = 40 μm/s and *a*_p_ = 12 μm; (**a**) surface microstructure; (**b**) enlarged view; (**c**) protrusion; (**d**) local microstructure; (**e**) EDS of position C; (**f**) contrast of foreign elements.

**Figure 7 materials-13-04664-f007:**
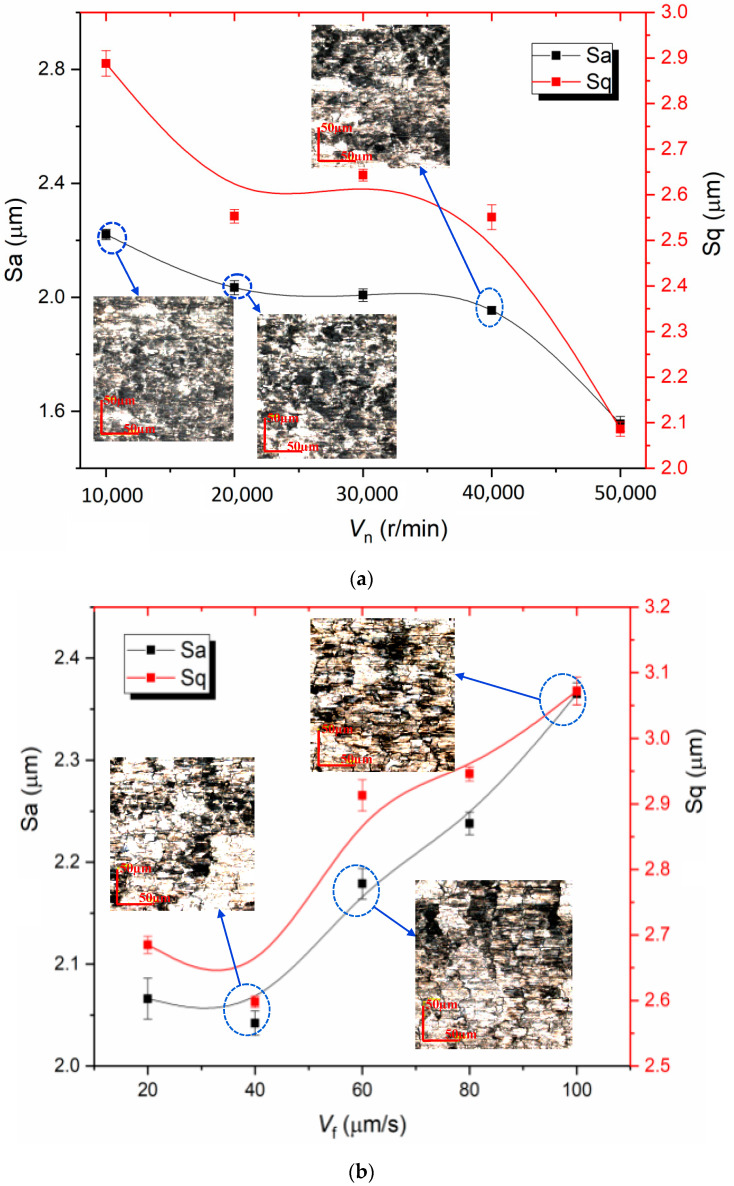
Effects of dry micro cutting parameters on surface roughness parameters of Al 6061 machined by MT-HMEM; (**a**) effects of *V*_n_ on *S*_a_ and *S*_q_; (**b**) effects of *V*_f_ on S_a_ and S_q_; (**c**) effects of *a*_p_ on *S*_a_ and *S*_q_; (**d**) contrast of Sa obtained with MT-HMEM and HMEM with various *a*_p_.

**Figure 8 materials-13-04664-f008:**
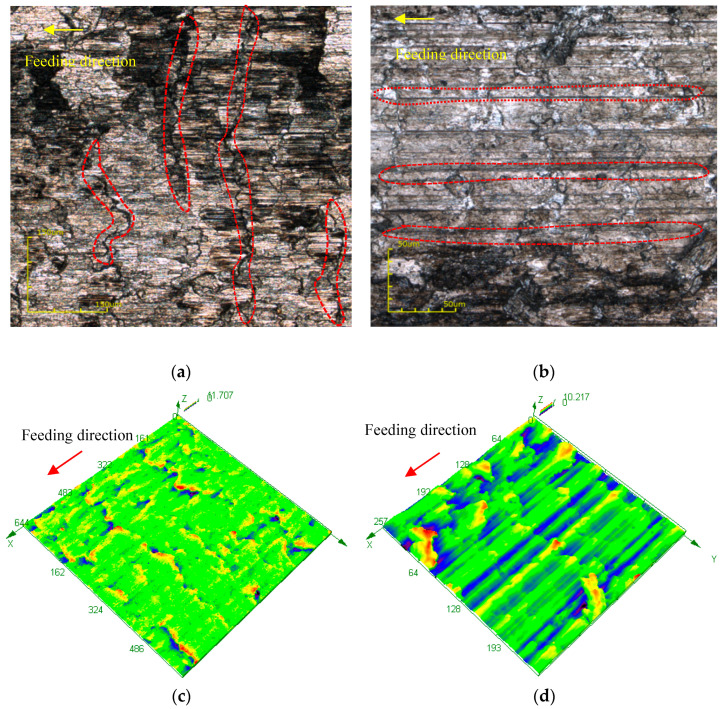
Contrast of dry micro cutting surfaces and Sa; (**a**) machined by MT- HMEM; (**b**) machined by HMEM; (**c**) 3D view machined by MT-HMEM; (**d**) 3D view machined by HMEM.

**Figure 9 materials-13-04664-f009:**
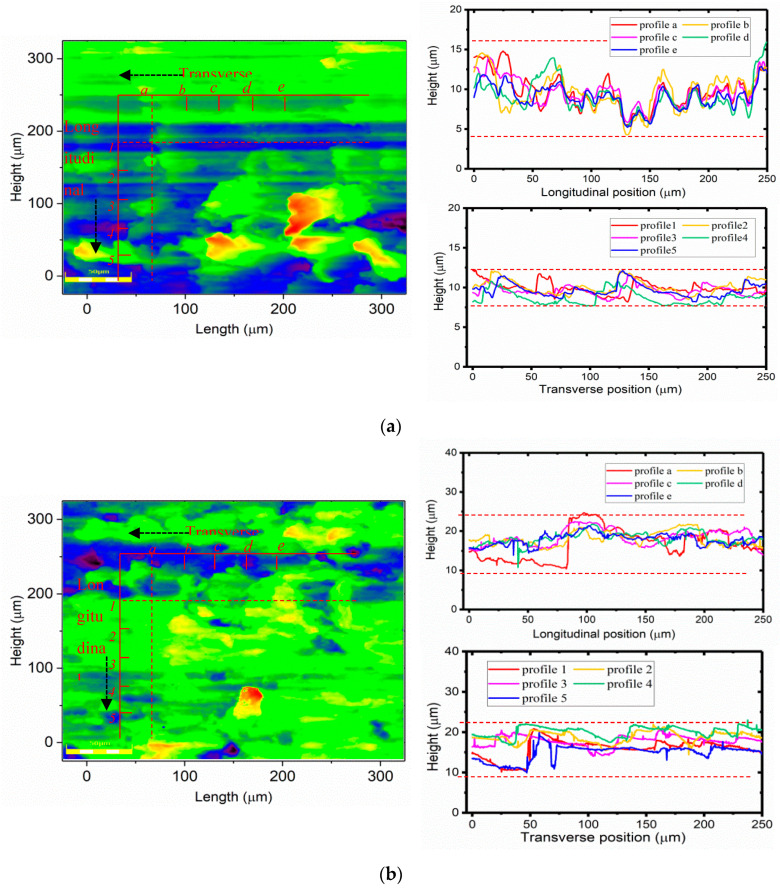
Dry micro cutting surface contour, (**a**) cross section contour obtained with HMEM, (**b**) cross section contour obtained with MT-HMEM.

**Figure 10 materials-13-04664-f010:**
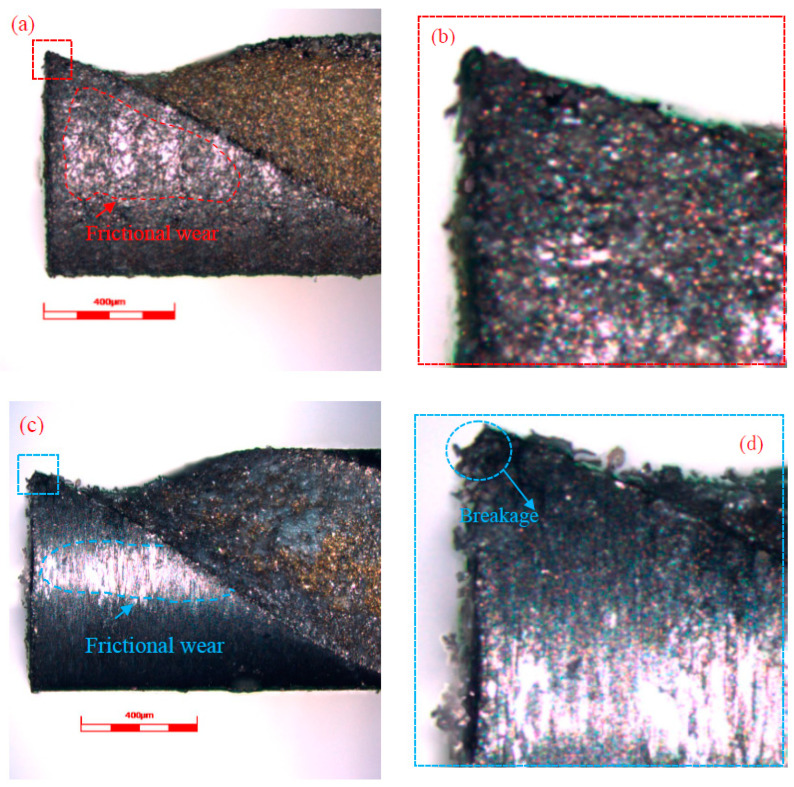
Tool wear of helical micro end mills with and without micro textures after dry micro cutting; (**a**) MT-HMEM; (**b**) tool tip of MT-HMEM; (**c**) HMEM; (**d**) tool tip of HMEM.

**Figure 11 materials-13-04664-f011:**
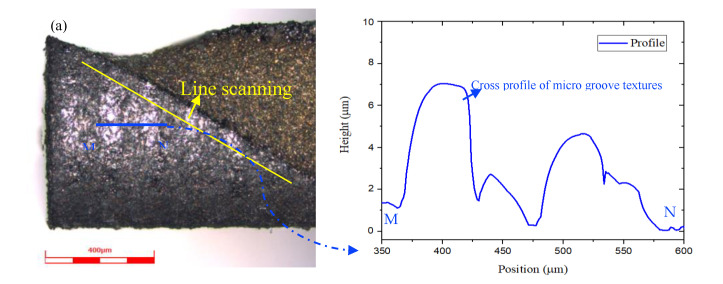
Line scanning analysis results of helical micro end mills with micro textures after dry micro cutting; (**a**) cross profile of micro textures; (**b**) C; (**c**) O; (**d**) Al; (**e**) W; (**f**) Co; (**g**) Cr.

**Table 1 materials-13-04664-t001:** Main parameters of LS-WEDM process for fabricating MT-HMEM.

Contents	Symbol	Unit	Rough	Trim	Helical	Micro Textures
Flushing pressure	*P*	kg/cm^2^	10	1	0.3	0.3
Peak current	*I*	A	320	120	180	120
Open voltage	*U*	V	170	100	100	85
Pulse on time	*T* _on_	μs	15	10	10	4
Rotating speed	*V* _n_	r/min	60	40	1	4
Wire tension	*F* _w_	N	12	18	15	15
Wire speed	*V* _s_	mm/s	30	30	30	30
Feeding speed	*f*	mm/min	0.5	1.2	4	0.7
Feeding amount	*a* _p_	μm	10	5	150	5

**Table 2 materials-13-04664-t002:** Chemical properties of Al 6061 [29].

Si	Cu	Fe	Mn	Mg	Ni	Pb	Zn	Ti	Sn	Cr	Al
0.43	0.24	0.43	0.139	0.802	<0.05	0.024	0.006	0.022	0.001	0.184	Balance

**Table 3 materials-13-04664-t003:** Experimental parameters for dry micro cutting.

No	*V*\*_n_* (r/min)	*V_f_* (μm/s)	*a_p_* (μm)
1	10,000, 20,000, 30,000, 40,000, 50,000	40	12
2	30,000	20, 40, 60, 80, 100	12
3	30,000	40	5, 8, 10, 12, 15

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
