# Peer review of "Experimental Investigation on Machinability of Aluminum Alloy during Dry Micro Cutting Process Using Helical Micro End Mills with Micro Textures"

_materials, 2020, doi:10.3390/ma13204664_

Round 1
Reviewer 1 Report
The paper is interesting, with a large number of illustrations. However, on some illustrations there are no reference dimensions or they are illegible. Fig. 8 a, b, c.
Fig. 9 a, b, c, d. Under Fig. 8 – minor errors in description.
The built-up edge phenomenon often occurs during the machining of aluminium alloys. Have the authors analysed this issue?
In Section 3.4. “Evaluation of tool wear of MT-HMEM after dry micro cutting”, the authors describe the wear of MT-HMEM tool. Can the authors specify the life of this tool?
Author Response
Dear Reviewers:
Thank you for your letter and for the reviewers’ comments concerning about our manuscript entitled “Experimental investigation on machinability of aluminum alloy during dry micro cutting process using helical micro end mills with micro textures". Those comments are all valuable and very helpful for revising and improving our paper, as well as important guiding significance to our researches. We have studied comments carefully and have made correction which we hope meet with approval. The revised portion is marked in blue color in the paper.
Sincerely yours,
Yao Sun
Reviewer1
- (Reviewer Comment) The paper is interesting, with a large number of illustrations. However, on some illustrations there are no reference dimensions or they are illegible. Fig. 8 a, b, c.
Response: Thanks for the reviewer’s kind advice. We have corrected accordingly and the scale dimensions of Fig.8 a, b, c are clear as displayed in our revised manuscript.
- (Reviewer Comment) Fig. 9 a, b, c, d. Under Fig. 8 – minor errors in description.
Response: Thanks for the reviewer’s kind advice. We have modified the minor errors in description for Fig. 9 a, b, c, d. Under Fig. 8 as displayed in our revised manuscript.
- (Reviewer Comment) The built-up edge phenomenon often occurs during the machining of aluminium alloys. Have the authors analysed this issue?
Response: Thanks for the reviewer’s kind advice. In this study, we mainly focused on fabricating helical micro end mills with micro texture and its machinability of micro cutting process, so we did not analyze the issue about the built-up edge. We thanks for
the reviewer’s insightful suggestions, and we will pay attention to study the built-up edge phenomenon in future work.
- (Reviewer Comment) In Section 3.4. “Evaluation of tool wear of MT-HMEM after dry micro cutting”, the authors describe the wear of MT-HMEM tool. Can the authors specify the life of this tool?
Response: Thanks for the reviewer’s kind advice. In micro cutting process, the micro tool interacts with the workpiece, and as the workpiece material is continuously removed, the diameter of micro cutting tool will be decreased and the tool tip are badly worn, which has a negative impact on the machining quality of the workpiece. When the tool is badly worn and cannot meet the quality of the workpiece, and the micro cutting process cannot be continued, so the tool failure is determined and the life of this tool is defined. As for the micro end mills with micro texture proposed in this study, when the micro textures have been grinded flat or the micro tool tip broke, we defined this micro tool failure. Based on the experimental results, when the cutting length is 15mm, the mean height of micro textures is reduced from 8μm to 6μm; when the cutting length is 40mm, the mean height of micro textures are almost flat, and the life of this tool is about 16min.

Reviewer 2 Report
The paper is on the micro cutting of aluminium alloy 6061 using helical mills that have micro textural patterns fabricated on them. The aim is to study the effect of machining parameters and the fabricated groves on 3D roughness metrics Sa and Sq.
- Line 24 “textures do not deteriorate” there is something missing in this sentence
- The abstract can be improving by adding more specific findings about the roughness parameters and what was found.
- Line 24-25 is self-repeating, consider rewriting it by clearing stating what was achieved in terms of surface roughness.
- Line 31 needs rewriting and does not read well “the micro cutting is not the simple…”
- English language is really poor and paper requires extensive English editing. Most sentences do not read well or clear and sometimes are difficult to understand.
- Authors should mention what each acronym stands for first time it shows in the paper. For example MQL: minimum quantity lubrication…
- Line 72 “from the literature, there are many studies paid attention...” the sentence does not read well. Consider revising.
- Line 94 “then the helical blades are obtained”
- In table 1, why were those certain parameters for micro texturing used for fabricating the grooves
- Remove and so on in line 106.
- How many milling tools were made which had the micro textures and how many without any textures? The authors need to add more details about the tools number and if all the textured tools were similar?
- Regarding the fabrication of micro texture on the tools, why this specific form of micro groves was applied on the surface of the milling tools. For example why not adding a sharp triangle like groves instead. Also the distance between the grooves, what was the rationale behind that?
- Did the authors validate the effect of groove density per unit of length on the surface roughness?
- Equation 3, if the author are using it from another reference/source then it should be referenced properly in the paper.
- The authors should remove the word dry in line 161 since they already mentioned the tests were carried out under dry environment.
- How many tools were used to confirm the repeatability of the experiments?
- There was no mention of tool wear analysis in the abstract, the authors should indicate that there.
- Line 270 needs revising “appear serious frictional…”
- Figure 11 is not clear and difficult to see the wear on the surface of the tools.
- Line 284 and 285 needs rephrasing.
- Figure 12 is not clear, is it showing heights or values of concentration of different elements? Show the unit in y axis.
Author Response
RE: materials-956453
Dear Reviewers:
Thank you for your letter and for the reviewers’ comments concerning about our manuscript entitled “Experimental investigation on machinability of aluminum alloy during dry micro cutting process using helical micro end mills with micro textures". Those comments are all valuable and very helpful for revising and improving our paper, as well as important guiding significance to our researches. We have studied comments carefully and have made correction which we hope meet with approval. The revised portion is marked in blue color in the paper.
Sincerely yours,
Yao Sun
Reviewer2
The paper is on the micro cutting of aluminium alloy 6061 using helical mills that have micro textural patterns fabricated on them. The aim is to study the effect of machining parameters and the fabricated groves on 3D roughness metrics Sa and Sq.
- (Reviewer Comment) Line 24 “textures do not deteriorate” there is something missing in this sentence
Response: Thanks for the reviewer’s kind advice. We have corrected accordingly as displayed in our revised manuscript.
- (Reviewer Comment) The abstract can be improving by adding more specific findings about the roughness parameters and what was found.
Response: Thanks for the reviewer’s kind advice. We have added specific roughness parameters in the abstract part as displayed in the revised manuscript.
- (Reviewer Comment) Line 24-25 is self-repeating, consider rewriting it by clearing stating what was achieved in terms of surface roughness.
Response: Thanks for the reviewer’s kind advice. We have added specific roughness parameters in the abstract part.
- (Reviewer Comment) Line 31 needs rewriting and does not read well “the micro cutting is not the simple…”
Response: Thanks for the reviewer’s kind advice. We have corrected the sentence to
be “But the micro cutting process has its own characteristics and rules, which is different from traditional cutting process” as displayed in our revised manuscript.
- (Reviewer Comment) English language is really poor and paper requires extensive English editing. Most sentences do not read well or clear and sometimes are difficult to understand.
Response: Thanks for the reviewer’s kind advice. This paper has been checked carefully and the English of the paper has been improved by native proof-reader.
- (Reviewer Comment) Authors should mention what each acronym stands for first time it shows in the paper. For example MQL: minimum quantity lubrication…
Response: Thanks for the reviewer’s kind advice. We have corrected accordingly as displayed in our revised manuscript.
- (Reviewer Comment) Line 72 “from the literature, there are many studies paid attention...” the sentence does not read well. Consider revising.
Response: Thanks for the reviewer’s kind advice. We have corrected the sentence to
be “From literature, many studies paid attention to fabricate different micro textural patterns on the rake or rear surface of turning tool, conventional size milling cutter, etc.,” as displayed in our revised manuscript.
- (Reviewer Comment) Line 94 “then the helical blades areobtained”
Response: Thanks for the reviewer’s kind advice. We have corrected accordingly as displayed in our revised manuscript.
- (Reviewer Comment) In table 1, why were those certain parameters for micro texturing used for fabricating the grooves
Response: Thanks for the reviewer’s kind advice. The micro textures located on the rear surface of micro end mills refer to micro grooves. For obtaining the regular array of micro grooves, the machining parameters should be fixed.
- (Reviewer Comment) Remove and so on in line 106.
Response: Thanks for the reviewer’s kind advice. We have removed “and so on” as displayed in the revised manuscript.
- (Reviewer Comment) How many milling tools were made which had the micro textures and how many without any textures? The authors need to add more details about the tools number and if all the textured tools were similar?
Response: Thanks for the reviewer’s kind advice. In this study, we only fabricate four micro end mills can be classified into two groups of MT-HMEM (helical micro end mills with micro texture) and HMEM (helical micro end mills).We have added the details about the tools number in the “Materials and Methods” part as displayed in the revised manuscript.
- (Reviewer Comment) Regarding the fabrication of micro texture on the tools, why this specific form of micro groves was applied on the surface of the milling tools. For example why not adding a sharp triangle like groves instead. Also the distance between the grooves, what was the rationale behind that?
Response: Thanks for the reviewer’s kind advice. We designed this micro grooves on the surface of the milling tools mainly based on the structure of pangolin scales and the corbicula shell as shown in Figure 1. The outer surface of pangolin scales has certain curvature change, and there are ribbed geometry and non-smooth structures in millimeter scale, and it is very strong and wear-resisting. Corbiculas also have a millimetre-scale ridged geometric structure distributed on the outer surface of their shells, which makes them capable to subject to abrasive wear of sea sand for a long time, so it also has excellent abrasive wear resistance. In this study, according to the surface morphology of corbicula shell, it can be found that it is consist of ribbed geometries. Therefore, the rib shape is reasonably simplified to the geometric features of sinusoidal function. The size and spacing of the crest and trough of the ripple shape are different as shown in Figure 2. And this is the reason of applying this kind of micro textures patterns on the surface of micro end mills.
(a) pangolin scaly (b) corbicula shell
Figure1 pangolin scaly and corbicula shell
Figure 2 The sectional shape of ribbed form
- (Reviewer Comment) Did the authors validate the effect of groove density per unit of length on the surface roughness?
Response: Thanks for the reviewer’s kind advice. In this study, we mainly focused on fabricating helical micro end mills with micro texture and the surface quality obtained with helical micro end mills with or without micro texture is compared, and we did not analyze the effect of groove density per unit of length on the surface roughness. We thanks for the reviewer’s insightful suggestions and we will pay attention to study the effects of micro textures patterns and their geometric parameters on surface roughness in future work.
- (Reviewer Comment) Equation 3, if the author are using it from another reference/ source then it should be referenced properly in the paper.
Response: Thanks for the reviewer’s kind advice. Based on the geometric relationship in the Figure.4, and the Eq.(3) can be deduced. Relevant explanations have been added into the revised manuscript.
- (Reviewer Comment) The authors should remove the word dry in line 161 since they already mentioned the tests were carried out under dry environment.
Response: Thanks for the reviewer’s kind advice. We have removed the word dry in line 161 as shown in the revised manuscript.
- (Reviewer Comment) How many tools were used to confirm the repeatability of the experiments?
Response: Thanks for the reviewer’s kind advice. Three micro end mills were used to confirm the repeatability of the experiments.
- (Reviewer Comment) There was no mention of tool wear analysis in the abstract, the authors should indicate that there.
Response: Thanks for the reviewer’s kind advice. We have added the relevant contents about tool wear of “The tool wear of helical micro end mills with micro textures involved in dry micro cutting process of Al 6061is rear frictional wear, oxidation wear and diffusion wear” into the abstract part as displayed in the revised manuscript.
- (Reviewer Comment) Line 270 needs revising “appear serious frictional…”
Response: Thanks for the reviewer’s kind advice. The “appear serious frictional…” has been changed to “appear frictional…” as shown in the revised manuscript.
- (Reviewer Comment) Figure 11 is not clear and difficult to see the wear on the surface of the tools.
Response: Thanks for the reviewer’s kind advice. Figure11 has been enlarged for expressing clearly, and the wear on the surface of the tools are marked as shown in the revised manuscript.
- (Reviewer Comment) Line 284 and 285 needs rephrasing.
Response: Thanks for the reviewer’s kind advice. We have rephrased the Line 284 and 285 as presented in the revised manuscript.
- (Reviewer Comment) Figure 12 is not clear, is it showing heights or values of concentration of different elements? Show the unit in y axis.
Response: Thanks for the reviewer’s kind advice. It means values of concentration of different elements and the unit in y axis is added into the revised manuscript.
Reviewer 3 Report
In this work, the authors implemented an experimental investigation on machinability of the aluminum alloy during the dry micro-cutting process using helical micro end mills with microtextures. The research appears to be efficiently done and appropriately reported, however, the standard of English is acceptable only needs few corrections. Nevertheless, there some questions and corrections that must be answered to improve and complete the document.
Line 39. When you use an abbreviation for the first time, you must indicate the meaning of it. Please, indicate in the text the meaning of “MQL”.
Lines 86-88. The authors claim that the EDM (they must indicate also the meaning of it) as “non-contact machining nature makes it more competent to fabricate micro cutting tools”. Please indicate the reasons that make EDM more competent for manufacturing the micro-cutting tools?
Lines 104-105, the authors evaluate and analysed the local surface morphology and surface element. However, they didn’t explain how they do it, or which technique and equipment were used to achieve the results presented in figures 3(e), 9 (c,d) and 10.
Line 113. In this line, the authors describe the meaning of the equation (1) parameters. However, the “k” parameter does not appear in the equation, so, there is any mistake: misses this parameter in the equation? The “k” could be changed by another symbol? The authors must correct this mistake. After the symbol “=” (equal), the authors wrote “0.1,2 , …”, I think that there is a mistake again and the correct statement is “0, 1, 2, …”, please verify this detail.
Line 258. The authors claimed, “… such a small amount of tool wear will cause…”. Can you quantify the advective "small"?
Lines 260-261. The authors observed in Figure 11 that in the HMEM occurred the breakage of tool nose while in the MT-HMEM did not appear this phenomenon. This happened for all experiments? How many tools did you tested?
Author Response
RE: materials-956453
Dear Reviewers:
Thank you for your letter and for the reviewers’ comments concerning about our manuscript entitled “Experimental investigation on machinability of aluminum alloy during dry micro cutting process using helical micro end mills with micro textures". Those comments are all valuable and very helpful for revising and improving our paper, as well as important guiding significance to our researches. We have studied comments carefully and have made correction which we hope meet with approval. The revised portion is marked in blue color in the paper.
Sincerely yours,
Yao Sun
Reviewer 3
In this work, the authors implemented an experimental investigation on machinability of the aluminum alloy during the dry micro-cutting process using helical micro end mills with micro textures. The research appears to be efficiently done and appropriately reported, however, the standard of English is acceptable only needs few corrections. Nevertheless, there some questions and corrections that must be answered to improve and complete the document.
- (Reviewer Comment) Line 39. When you use an abbreviation for the first time, you must indicate the meaning of it. Please, indicate in the text the meaning of “MQL”.
Response: Thanks for the reviewer’s kind advice. The MQL represents the minimum quantity lubrication, and we have corrected accordingly as displayed in our revised manuscript.
- (Reviewer Comment) Lines 86-88. The authors claim that the EDM (they must indicate also the meaning of it) as “non-contact machining nature makes it more competent to fabricate micro cutting tools”. Please indicate the reasons that make EDM more competent for manufacturing the micro-cutting tools?
Response: Thanks for the reviewer’s kind advice. This is because EDM method will not have direct physical contact with micro cutting tools, which is different from turning or grinding methods, so the micro cutting tools will not be easy to break or bend. Thus, the EDM method is more suitable for fabricating micro cutting tools. The relevant contents have been added into the revised manuscript.
- (Reviewer Comment) Lines 104-105, the authors evaluate and analysed the local surface morphology and surface element. However, they didn’t explain how they do it, or which technique and equipment were used to achieve the results presented in figures 3(e), 9 (c, d) and 10.
Response: Thanks for the reviewer’s kind advice. The dry micro cutting experiments are performed on JX-1A precision micro-machining tool as shown in Figure.1
- (Reviewer Comment) Line 113. In this line, the authors describe the meaning of the equation (1) parameters. However, the “k” parameter does not appear in the equation, so, there is any mistake: misses this parameter in the equation? The “k” could be changed by another symbol? The authors must correct this mistake. After the symbol “=” (equal), the authors wrote “0.1,2 , …”, I think that there is a mistake again and the correct statement is “0, 1, 2, …”, please verify this detail.
Response: Thanks for the reviewer’s kind advice. We have modified the “k= 0.1, 2 , …” to be “x= 0,1,2 ,…” as displayed in the revised manuscript.
- (Reviewer Comment) Line 258. The authors claimed, “… such a small amount of tool wear will cause…”. Can you quantify the advective "small"?
Response: Thanks for the reviewer’s kind advice. It is difficult to quantify the “small”, so we have rephrased this sentence for the sake of rigor, this sentence has been changed to “the tool wear will cause the large machining dimensional error and seriously affect the micro cutting force…..” as displayed in the revised manuscript.
- (Reviewer Comment) Lines 260-261. The authors observed in Figure 11 that in the HMEM occurred the breakage of tool nose while in the MT-HMEM did not appear this phenomenon. This happened for all experiments? How many tools did you tested?
Response: Thanks for the reviewer’s kind advice. We have tested two groups of HMEM and MT-HMEM, and experimental results indicated that HMEM occurred the breakage of tool nose.

Round 2
Reviewer 2 Report
Check line 27 "process of Al 6061is rear frictional wear"
Line 75 Replace But with However
paper then can be accetped after thorough English checking
Author Response
- (Reviewer Comment) Check line 27 "process of Al 6061is rear frictional wear"
Response: Thanks for the reviewer’s kind advice. We have corrected accordingly as displayed in our revised manuscript.
- (Reviewer Comment) Line 75 Replace But with However
Response: Thanks for the reviewer’s kind advice. We have corrected accordingly as displayed in our revised manuscript.
3.(Reviewer Comment) paper then can be accetped after thorough English checking.
Response: Thanks for the reviewer’s kind advice. This paper has been checked carefully and the English of the paper has been improved by native proof-reader.
